# Hybrid Inorganic Organic PSF/Hap Dual-Layer Hollow Fibre Membrane for the Treatment of Lead Contaminated Water

**DOI:** 10.3390/membranes13020170

**Published:** 2023-01-30

**Authors:** Sumarni Mansur, Mohd Hafiz Dzarfan Othman, Nurul Jannah Ismail, Siti Hamimah Sheikh Abdul Kadir, Mohd Hafiz Puteh, Huda Abdullah, Juhana Jaafar, Mukhlis A. Rahman, Tutuk Djoko Kusworo, Ahmad Fauzi Ismail, Abdul Latif Ahmad

**Affiliations:** 1Advanced Membrane Technology Research Centre (AMTEC), Faculty of Chemical and Energy Engineering, Universiti Teknologi Malaysia, Johor Bahru 81310, Johor, Malaysia; 2Laboratory and Forensics (I-PPerForM), Institute of Pathology, Faculty of Medicine, Universiti Teknologi Mara (UiTM), Cawangan Selangor, Sungai Buloh 47000, Selangor, Malaysia; 3School of Civil Engineering, Faculty of Engineering, Universiti Teknologi Malaysia, Skudai 81310, Johor, Malaysia; 4Department of Electrical, Electronic & Systems Engineering, Faculty of Engineering & Built Environment, The National University of Malaysia, Bangi 43600, Selangor, Malaysia; 5Department of Chemical Engineering, Faculty of Engineering Diponegoro University, Semarang 50275, Indonesia; 6School of Chemical Engineering, Universiti Sains Malaysia, Engineering Campus, Nibong Tebal 14300, Pulau Pinang, Malaysia

**Keywords:** lead adsorption studies, waste cockle shell, hydroxyapatite, dual-layer hollow fibre membrane, lead removal

## Abstract

Lead (Pb) exposure can be harmful to public health, especially through drinking water. One of the promising treatment methods for lead contaminated water is the adsorption-filtration method. To ensure the cost-effectiveness of the process, naturally derived adsorbent shall be utilised. In this study, hydroxyapatite particles, Ca_10_(PO_4_)_6_(OH)_2_ (HAP) derived from waste cockle shell, were incorporated into the outer layer of polysulfone/HAP (PSf/HAP) dual-layer hollow fibre (DLHF) membrane to enhance the removal of lead from the water source due to its hydrophilic nature and excellent adsorption capacity. The PSf/HAP DLHF membranes at different HAP loadings in the outer layer (0, 10, 20, 30 and 40 wt%) were fabricated via the co-extrusion phase inversion technique. The performance of the DLHF membranes was evaluated in terms of pure water flux, permeability and adsorption capacity towards lead. The results indicated that the HAP was successfully incorporated into the outer layer of the membrane, as visibly confirmed by microscopic analysis. The trend was towards an increase in pure water flux, permeability and lead adsorption capacity as the HAP loading increased to the optimum loading of 30 wt%. The optimized DLHF membrane displayed a reduced water contact angle by 95%, indicating its improved surface hydrophilicity, which positively affects the pure water flux and permeability of the membrane. Furthermore, the DLHF membrane possessed the highest lead adsorption capacity, 141.2 mg/g. The development of a hybrid inorganic–organic DLHF membrane via the incorporation of the naturally derived HAP in the outer layer is a cost-effective approach to treat lead contaminated water.

## 1. Introduction

Lead service lines can be defined as pipes that are made up of lead, which is used in potable water distribution to connect the main drinking water line to a household [1]. Lead service lines have been restricted globally due to the risk of lead-contaminated water induced by leaded pipes [2]. It is found that lead service lines are the largest source of lead in drinking water. Plumbing components containing lead can deteriorate and release lead into drinking water, especially in situations where the water has a high acidity or low mineral content that corrodes pipes and fixtures [3]. The most common sources of lead in drinking water are lead pipes, faucets, and fixtures. Although legislation and policies have been implemented and enforced, lead exposure is still a major concern for public health [4,5]. Based on data from the United States Environmental Protection Agency (EPA), it can be estimated that there are currently 6–10 million lead service lines across the United States [4]. High replacement and installation costs have been one of the main factors in the existence of lead service lines despite strict law enforcement regarding the use of leaded pipes. Corrosion control helps in reducing the risk of lead contamination in water. Lead or other metals can be stopped from leaching into the water by using anti-corrosion chemicals [6]. Lead pipes are coated on the interior with a thin, protective coating by corrosion inhibitors such zinc orthophosphate, which prevents leaching and flaking. However, this method did not solve the problem of lead contamination in drinking water. The World Health Organization (WHO) has established the guideline that lead concentrations in drinking water should not to exceed 0.01 mg/L [7]. Thus, it is important to find an effective method to treat lead-contaminated water, especially for drinking.

The challenges in removing lead from wastewater is that it is highly persistent, meaning lead is in its elemental form, and hence cannot be degraded into a less dangerous form [8]. In addition, lead can be highly toxic and harmful to living organisms even at trace levels [7]. Heavy metals such as lead can be removed from wastewater through physical and chemical methods. Physical methods include ion exchange, coagulation, ultrafiltration, adsorption, membrane filtration, and flocculation [9], while chemical methods include chemical precipitation, neutralization, electrochemical treatment, and solvent extraction. Evidently, there are various competing methods for removing lead. Therefore, the key principle is to develop a method that is low cost and more reliable.

The development of highly efficient and cost-effective technologies for the clean treatment of these wastewater streams has been widely explored. Membrane separation exhibits more advantages over conventional separation techniques in the purification of water since it is easy to use, effective, relatively energy efficient, and operates at ambient temperature [10,11]. Ultrafiltration (UF), one of the pressure-driven membrane technologies, has been employed in industries and residential water treatment applications to remove dissolved macromolecules and finer colloids from aqueous streams. Most of the UF commercial membranes are synthesized using organic polymers and their derivatives [12]. The performance of the UF polymeric membrane is greatly hindered by fouling processes, which raise the cost of operation and maintenance. Polysulfone (PSf) has been widely used in the production of the UF membrane due to its physico-chemical properties which include good thermal, chemical resistance and mechanical properties [13,14]. However, PSf is hydrophobic by nature and is easily fouled [15]. Numerous studies have shown that hydrophilic membrane modification can greatly reduce flow resistance by preventing foulants from adhering to the membrane surface, hence enhancing the antifouling properties of the membrane.

Due to its chemical and structural similarity to the mineral component of bones, hydroxyapatite (HAP) is widely used in biomedical areas [16,17]. The non-toxicity, biocompatibility, and chemical stability of this inorganic substance have led to its reputation as a flexible substance. Additionally, HAP has excellent adsorption abilities. On the surface of HAP are two separate binding sites, designated as C and P sites. Positively charged calcium ions form C sites, which preferentially adsorb acid molecules, whereas negatively charged phosphate groups form P sites, which primarily adsorb basic compounds [18]. Furthermore, the presence of many hydroxyl groups in HAP makes them hydrophilic by nature, which may enhance the UF membranes’ antifouling capabilities. [19]. Cockle shells (Anadara granosa) are the bio-waste of the Mollusca phylum and can be found abundantly among other seashell waste along the shores of any islands. Normally, these shells are used for ornaments and tourists buy them as souvenirs. It has been reported that cockle shell has high potential to be used as calcium precursor for HAP synthesis. This is because cockle shells are 89–99% calcium carbonate.

In this study, HAP was extracted using cockle shell waste as the starting material through a simple calcination process. The extracted HAP was then incorporated into the outer layer of PSf/HAP dual-layer hollow fibre (DLHF) membrane. The HAP was concentrated on the outer layer of the membrane as the adsorption–filtration process mostly occurred on the outer layer (out-to-in crossflow system). The physico-chemical, permeability performance and adsorption efficiency of the synthesised HAP powder and fabricated PSf/HAP DLHF membrane were thoroughly investigated and compared to pristine PSf membrane.

## 2. Experimental

### 2.1. Materials

HAP powder was synthesised from waste cockle shells that was obtained from a market in Johor. Polysulfone (PSF, udel-1700, mw 69,500 g/mol) was obtained from Solvay Advanced Polymers, Neder-Over-Heembeek, Belgium. N-methyl-2-pyrrolidine (NMP) and hydrophilic pore former, polyvinylpyrrolidone (PVP) were purchased from Sigma-Aldrich, St. Louis, MO, USA. 1000 ppm of lead stock solution was obtained from Sigma-Aldrich (St. Louis, MO, USA).

### 2.2. Synthesis of Hydroxyapatite Powder from Waste Cockle Shell

Tap water was used to wash the waste cockle shells obtained from the market to remove the dirt and attached tissue on the cockle shells. Prior to calcination, the cockle shells were dried in an oven at 100 °C overnight. Next, the shells were crushed and calcined at 800 °C for 3 h in a muffle furnace with a heating rate of 10 °C/min. The furnace’s temperature was allowed to decrease at a rate of 10 °C/min until it reached room temperature. After calcination, the shells were ground and milled into powders using a planetary ball mill. The obtained powder was sieved using a 36 μm sieve for 20 min to obtained fine HAP powder.

### 2.3. HAP Powder Characterisation

The synthesised HAP powder was characterised based on Fourier-transform infrared (FTIR), X-ray powder diffractometer (XRD), scanning electron microscopic (SEM), energy dispersive X-ray (EDX) analysis and Brunauer–Emmet–Teller (BET) analysis. Functional groups in the powders were identified using an FTIR system with a diamond attenuated total reflectance (ATR) system (Thermo Fisher Nicolet iS5, Waltham, MA, USA) at a range of 400 to 4000 cm^−1^. The KBr pellet technique was used to prepare the sample. The phases of the powders were analysed using XRD (Rigaku MiniFlex II, Tokyo, Japan) with step size 0.02° at 1 s step time. SEM analysis with EDX (Hitachi TM3030 Plus, Hitachi, Japan) was used to observe and determine the morphology and elemental analysis of the powders. To avoid charging during the analysis process, the samples were sputter-coated with platinum.

### 2.4. Fabrication of Dual-Layer Hollow Fibre Membrane

Before making the dope solutions, PSf, PVP, and synthesised HAP powder was dried at 50 °C for the entire night. M0 represents neat, single-layer hollow fibre (SLHF) membrane, while M10–M40 represent DLHF membrane with increase weight percent (wt%) of HAP powder on the outer layer of the hollow fibre membrane (Table 1). The HAP powder was stirred until well dispersed in pre-calculated NMP solvent. Next, PVP was gradually added into the solution and stirred for another 2 h. Lastly, PSf was added gradually. The solution was stirred overnight until all the particles completely dissolved and homogenized. The dope solution was stirred using an overhead mechanical stirrer.

The DLHF membranes were spun using co-extrusion spinning technique using a custom-made triple orifice spinneret. The dope extrusion rate (DER) for inner dope solution was maintained at 1:1 ratio with the bore fluid flow rate (BFFR) where the DER and BFFR was run at 8 mL/min. Meanwhile, the DER of the outer dope solution was run at 1 mL/min. To remove any remaining NMP solution, the fabricated DLHF membranes were immersed under flowing tap water for 48 h. In order to prevent the membrane structure from collapsing, the membranes were then treated with 10% glycerol and submerged for 24 h. The DLHF membranes underwent air drying at room temperature prior to storage.

### 2.5. Membrane Characterisation

The whole fabricated membrane was analysed based on based on SEM images, EDX analysis, and inner surface contact angle goniometer.

The SEM (Hitachi, USA) was used to observe the DLHF membranes’ cross-sectional and surface morphology. An approach called freeze fracturing was used to prepare the membrane’s cross-sectional surface. To achieve a smooth cutting of the cross-sectional surface, the membrane was snapped while the membrane was immersed in liquid nitrogen for few seconds. The membrane’s surface was prepared by cutting a 1 cm-long piece of the membrane. The prepared samples were placed on stab. The sample was sputter-coated with gold before the SEM analysis to reduce noise.

The hollow fibre membrane’s elemental composition was identified and measured using EDX. The sample was assessed at accelerating voltage of 15 kV. The mapping and spectral point pictures of each sample were viewed and analysed at a magnification of 250×. To get an average value, the analysis was performed three times to determine carbon (C), oxygen (O), sulphur (S), and silicon (Si) atomic concentration.

Surface hydrophilicity property of the membrane was analysed using a contact angle goniometer (Dataphysics, Germany). A strand of membrane fibre was set at a designated stage, where 0.2 μL water droplet was dropped on the surface of the membrane at fixed 1.0 μL/s of dosing rate using the sessile drop method. The water droplet’s shadow on the membrane was observed and the contact angle value of the water droplet on the surface of the membrane was recorded. Ten measurements of the contact angle were taken for each fibre to ensure a valid measurement.

### 2.6. Membrane Permeability Analysis

A crossflow ultrafiltration experiment setup was used to examine the membrane’s permeability and flux. Tap water was used as the feed and allowed to pass through membrane module. A stabilizing test was carried out prior to the permeability analysis for 10 min at 1.0 bar. Next, the permeate was collected and measured for every 10 min at a fixed pressure of 0.5 bar. To obtain an average result, the testing was conducted three times. The Equation (1) was used to determine the membrane’s permeability.
(1)$$\mathrm{Permeability}\;(L/m^{2}\cdot h\cdot \mathrm{bar})=\frac{V}{A\times \Delta t\times P}$$
where *V* indicated the volume of permeate (L) over the area of the hollow fibres, *A* (m^2^), the time taken, *t* (h), and the pressure, *P* (bar).

### 2.7. Adsorption Analysis

The adsorption characteristic of HAP was determined by varying the initial concentrations of arsenic in water (0–1000 μg/L). For this, 0.1 g of adsorbent was added into conical flask (50 mL) containing the prepared lead solution (10 mL). The mixture was then stirred in a conical flask for 24 h at room temperature at 160 rpm using an orbital shaker (Miulab, GS-20, China). Atomic absorption spectroscopy (AAS, Agilent, USA) was used to detect the solution after filtering it with a filter syringe. Equation (2) was used to determine the amount of p-cresol adsorbed per unit gramme of adsorbent [20,21].
(2)$$q_{e}=\frac{(C_{o}-C_{e})\;V}{m}$$
where *q_e_* is the amount of solute adsorbed at equilibrium per adsorbent weight (mg/g), *C_o_* and *C_e_* are the initial concentration and the equilibrium concentration of solute, respectively (mg/L), *V* is the volume of the solution (L) and *m* is the initial mass of the adsorbent (g).

Langmuir and Freundlich isotherm models were used to further examine the mechanism of lead adsorption onto HAP powder. The linear equations provided in Equations (3) and (4) were used to describe the Langmuir and Freundlich models (4), respectively [21,22]:(3)$$\frac{C_{e}}{q_{e}}=\frac{1}{Q_{max}b}+\frac{C_{e}}{Q_{max}}$$
(4)$$\mathrm{Ln}q_{e}=\ln K_{F}+\left(\frac{1}{n}\right)\ln C_{e}$$
where *q_e_* is the amount of lead adsorbed on HAP (mg/g), and *C_e_* is the equilibrium lead concentration(mg/L); *Q_max_* is maximum adsorption capacity (mg/g); *b* is the Langmuir constant related to binding energy of the sorption system (L/g). *K_F_* is the Freundlich constant (mg/g) and 1/*n* is the heterogeneity parameter.

### 2.8. Dynamic Adsorption

The membrane dynamic adsorption performance was tested through the same crossflow ultrafiltration experiment system. For this test, 100 ppm of lead was used as feed to test the adsorption capacity of the fabricated membrane in removing lead waste. The tested membrane was cut to 20 cm length and subjected to a pressure of 1.5 bar for 20 min prior the experiment to stabilize the flux of the fibre. A pressure of 0.5 bar was used to pressurize the permeate through the membrane and the permeate was collected at 15 min time interval for 2 h. The percentage of lead removal for each collection time was calculated based on the Equation (5).
(5)$$\%\;of\;removal=\frac{C_{o}-C_{e}}{C_{o}}\times 100\%$$
where the *C_o_* is the initial concentration of the feed and *C_e_* is the final concentration of the permeate.

## 3. Results and Discussion

### 3.1. Morphology and Chemical Characteristic of the Synthesize HAP Powder

From the SEM micrographs (Figure 1), the cockle shell powder was found to be irregular in shape and varied in size; this morphology of powder was believed to be HAP, as it was parallel in agreement with the studies by Dey et al. (2014) and Bogdanoviciene et al. (2006), where the produced HAP have irregular shapes.

Figure 2 shows the FTIR spectra of waste cockle shell in which possessed C-O band presented at 1417 cm^−1^ and 877 cm^−1^. Meanwhile, bands 1089 cm^−1^, 1028 cm^−1^, 962 cm^−1^, 602 cm^−1^ and 561 cm^−1^ corresponded with the phosphate band. Furthermore, a hydroxyl band can also be seen at 3347 cm^−1^. The two weak adsorption peaks at 2090 cm^−1^ and 2190 cm^−1^ correspond to the out-of-plane bending vibration of C-O carbonates. This finding was parallel to the finding by previous group [23]. Since FTIR analysis is meant to detect the functional group of compounds, this finding proved that HAP has been successfully synthesized from the ground cockle shell powder. Similar findings were also obtained by previous scholars (Dey et al., 2014; Bogdanoviciene et al., 2006).

The EDX analysis of cockle shell powder confirmed that there was a presence of HAP powder based on the measurement of the elemental composition (Ca, C, O and P content) as shown in Table 2. The result was compared with the previous studies as summarized in Table 2. These substances are consistent with the FTIR results in Figure 2, which show the presence of phosphate groups in the powder. The presence of carbon may result from the reaction’s absorption of carbon dioxide.

The XRD patterns of the cockle shell powder showed that HAP has a high crystallinity as shown in Figure 3. In order to analyse the XRD results, they were compared to the standard diffraction pattern of JCPDS No. 09-0432 in the 20° to 60° range. Based on the comparison, the crystalline phases of cockle shell powder revealed peaks that have the same intensity as the HAP standard, thus proving that HAP was successfully synthesized from the cockle shell powder.

### 3.2. Lead Removal Studies Based on HAP Powder Adsorption Capacity

#### 3.2.1. Effect of Initial Lead Concentration for Synthesize HAP Powder

Figure 4 presents the removal of lead by the synthesis HAP particle with respect to different initial concentrations (0–1000 mg/L) of lead. Because there are still binding sites for lead molecules, a sharp increase in lead absorption can be observed with increasing initial urea content, demonstrating the substance’s potent adsorption ability. In fact, the larger driving force caused by the higher concentration difference encourages adsorption at initial lead concentrations greater than 1000 mg/L.

#### 3.2.2. Effect of Contact Time for Synthesize HAP Powder

Figure 5 displays the findings of the study on the effect of contact time on lead adsorption by HAP particles. Within the first five minutes of coming into contact with HAP, the rate of urea adsorption was high, and more than 80% clearance had been accomplished. This initial fast adsorption was caused by the lengthy structure of HAP containing aldehyde groups. The adsorption rate started to become slower after 10 min of adsorption analysis. This slow adsorption rate was due to the attachment of lead on the HAP surface via temporary hydrogen bonding. Thus, lesser binding site availability for the adsorption to occur. After 1 h, the adsorption process had finally reached equilibrium., with95% removal (equivalent to 17.7 mg/g).

#### 3.2.3. Adsorption Isotherm

Table 3 shows the isotherm parameters of Langmuir and Freundlich models, together with respective correlation coefficients, R^2^ for lead adsorption. According to the table, the R^2^ value from the Langmuir model was greater than that from the Freundlich model, demonstrating that the Langmuir isotherm was the best to explain the process of adsorption between the HAP and lead molecules [21,22]. Based on Langmuir model, it can be estimated that lead was deposited only on the free surface of HAP throughout the adsorption process, creating a monolayer of lead. As the adsorbent surface is uniform and has the equivalent adsorptive sites, the same mechanism was applied to all lead molecules with no interaction between them.

#### 3.2.4. Adsorption Kinetics

The kinetic parameters and R2 values of the two kinetic models (pseudo-first order and pseudo-second order) are shown in Table 4. The lead adsorption employing HAP powder followed the former model, as shown by the correlation coefficient of the pseudo-second order kinetics being higher than that of the pseudo-first order kinetics. Therefore, it can be claimed that the adsorption nature belongs to the class of chemisorption, where the attraction between HAP and lead was caused by chemical bonding rather than by weak forces of physisorption. The results were in accordance with the Langmuir isotherm, given that chemisorption is connected to the development of a single layer of adsorbate. In general, chemisorption is extremely selective and requires a possible chemical bonding between the adsorbent and the adsorbate. The lead adsorption on the adsorptive sites of HAP powder in this instance was controlled via a chemical reaction process. The adsorption capacity result of HAP powder shows that HAP powder is a good adsorbent for lead removal. However, it presents a recovery issue as the adsorbate attaches to the adsorbate chemically. Thus, it needs to be immobilised in the substrate, where in this study, HAP powder was incorporated into the outer layer of the PSf DLHF membrane.

### 3.3. Morphology Study of the Fabricated Membrane

Based on the SEM images in Figure 6, hybrid inorganic organic PSf/HAP DLHF membranes were successfully fabricated. The neat PSf membrane, M0 produced dense skin layer at the innermost layer and thick sponge-like support layer. The support layer formed a porous finger-like structure, becoming a more sponge-like macrovoid layer, and then returned to a porous finger-like structure as it reached the edge of the membrane. The inner layer of the DHLF membrane shows the same structure as neat PSf membrane, M0. PSf/HAP DLHF membrane was produced by co-extrusion of PSf incorporated with different concentration of HAP dope solution on top of the first PSf nascent inner layer. As shown in Figure 7, a second layer formed on outermost of the membrane and shows no sign of delamination between the two layers. The thickness of the outer layer of the membranes were 69.3 μm, 79.7 μm, 95 μm and 120 μm for M10, M20, M30 and M40, respectively. As the concentration of HAP increases, the outer dope solution became more viscous. The high viscosity of the dope solution when extruded out from the spinneret slowed down the phase inversion, producing a thicker and denser outer layer of the DLHF membrane. Thus, the higher the percentage of HAP in the outer layer composition of the membrane, the thicker the outer layer formed. At 30 wt% of HAP concentration, the outer layer formed a few macrovoid and finger-like structures. The finger-like structure on the outer layers became more prominent as the concentration of HAP was increased to 40 wt%. As the phase inversion slows due to the high viscosity of the dope solution, the coagulation bath moves slower and becomes trapped in the nascent fibre, and thus formed the macrovoid and finger-like structure.

Figure 7 shows the outer surface of the fabricated PSf/HAP DLHF membrane. The neat PSf membrane, M0, presents a well-spread pore structure on the surface of the membrane. During the solvent/nonsolvent exchange mechanism in the phase inversion process, the pore former (PVP) leached out from the nascent fibre, hence producing pores. The HAP can be observed as agglomerate white particles that were spread across the membrane; they did not appear on the M0 surface, but were visible on the entire surface of the PSf/HAP DLHF membrane. As higher concentration of HAP becomes incorporated into the outer layer of the membrane, the HAP white particle becomes less visible on the surface of the membrane. Instead, the HAP white particle becomes embedded within the matrix of the membrane. The high viscosity of the dope solution hindered the flow of solvent exchange during phase inversion, thus creating a dense and thick structure. The HAP particle did not leach out from the membrane and trapped within the matrix of the membrane. The trapped HAP particle helps in enhancing the performances of the membrane in terms of permeability and adsorption.

The formation of dual layer of PSf/HAP DLHF membrane is hard to distinguish due to the similarity to the structure and colour of the fabricated membrane. Thus, to determine the second layer of the membrane that incorporated with HAP, elemental analysis was performed on the cross-sectional surface of the PSf/HAP DLHF membrane via EDX mapping and point ID. Figure 8 shows the EDX mapping images corresponding to carbon (C), sulphur (S), oxygen (O), calcium (Ca) and phosphorus (P). C and S elements were from the main component of PSf while Ca and P came from the major elements existing in HAP (Ca_10_(PO_4_)_6_(OH)_2_. Based on Figure 9, Ca and P elements only appeared on the DLHF membrane, whereas the elements were not seen on the neat PSf membrane, M0. The newly appearing second layer on the outer layer of the membrane observed through EDX mapping and point ID has proven the successful formation of a dual-layer hollow fibre membrane. This result also indicated that the hydrophilic HAP did not leach out of the membrane and was trapped within the matrix of the membrane.

### 3.4. Hydrophilicity/Hydrophobicity

The hydrophilicity/hydrophobicity of the membranes was determined using water contact angle measurement as plotted in Figure 9. The pristine PSf membrane, M0 with water contact angle of 75° dropped to 68.35ᵒ as 10 wt% of HAP was added on the outer surface of the membranes, M10. As the weight percentage of HAP increased, the membrane’s water contact angle gradually decreased, demonstrating the membranes’ improved surface hydrophilicity. Higher concentrations of OH^−^ and PO_4_^3−^ of HAP at the membrane surface allowed for greater hydrogen bonding interactions between the membrane and water molecules, which increases the capacity of the PSf/HAP DLHF membrane to absorb water and store more water molecules [19]. As seen in Figure 10, the M40 membrane has the highest contact angle measurement. High concentrations of HAP in the membrane cause a more compact structure within the matrix of the membrane, thus resulting in a more hydrophobic surface.

### 3.5. Membrane Flux and Permeability

Figure 10 presents the water flux and permeability performance of the fabricated PSf/HAP DLHF membrane. The permeability performance results agreed with the hydrophilicity results of the membranes. After the addition of 10% HAP in the outer layer of the membrane, the permeability of the membrane increases up to 82%. The hydrophilic property of the outer layer of the membrane enhances the water flux and permeability of the membrane. The higher the concentration of HAP incorporated into the outer layer of the membrane, the higher the flux and permeability of the membrane. However, at 40 wt% of HAP, the dope solution was too viscous, causing the membrane to form a very dense, compact, and thick structure, suppressing the flow of water across the membrane, hence reducing its permeability. This phenomenon was also demonstrated in a study by Kallem et. at. (2021), where an additive could affect the membrane formation if the concentration was too high, thus leading to relatively low flux for the membrane [19].

### 3.6. Dynamic Adsorption od PSf/HAP DLHF Membranes

M30 and M40 show the highest percentage of removal, as shown in Figure 11. The dense and thick outer layer of the DLHF membrane helps to adsorb more heavy metals. The higher the concentration of HAP powder in the outer layer of the DLHF membrane, the higher the removal efficiency of the membrane. Initially, M20 shows an increasing trend and starts to drop after 30 min of filtration. This happens may because of the chemisorption relation between lead and HAP. As the active sites were fully occupied, the adsorption site started to be enclosed, resulting in this low adsorption capacity after 30 min of filtration. This result shows that 20 wt% of HAP powder is not enough to sustain the adsorption of the lead for up to 2 h of dynamic adsorption–filtration. M0 and M10 show no lead removal occurring for 2 h of the dynamic adsorption experiment. Neither membrane has enough HAP powder for the adsorption process to occur. This phenomenon is in coherent with the study by Zaman et al., where a high concentration of HAP powder is needed to maintain the adsorption capacity of the membrane during filtration. The presence of HAP in membrane matrix creates a synergetic effect in imparting -OH polar end ties and improving the capillary network for adsorption to occur. Thus, this proves that a high concentration of HAP was needed on the outer layer for higher adsorption of heavy metal to occur.

## 4. Conclusions

The study focuses on the synthesis of a novel UF membrane material prepared by incorporation of HAP into the outer layer of PSf/HAP DLHF membrane. HAP powder was successfully synthesized from waste cockle shells. The synthesized HAP shows a good adsorption capacity towards lead with 180.8 mg/g of adsorption capacity. The synthesized HAP was then incorporated into PSf DLHF membrane. This membrane was to fabricate novel high flux hybrid inorganic–organic ultrafiltration membranes. The presence of HAP particles significantly increased the hydrophilicity of the hybrid PSf/HAP DLHF membrane. The permeability of all PSf/HAP DLHF membranes was higher compared to the pristine neat PSf membrane. The results showed that the 30 wt% HAP possessed the highest flux and adsorption capacity towards lead among the fabricated membrane series. Evidently, the results showed that the developed PSf/HAP DLHF membranes are acceptable for water treatment processes, specifically for lead-contaminated wastewater.

## Figures and Tables

**Figure 1 membranes-13-00170-f001:**
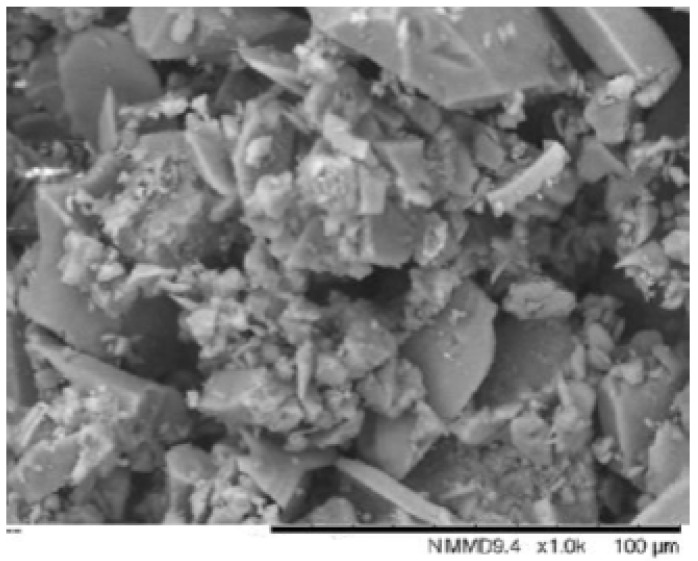
SEM image of cockle shell powder.

**Figure 2 membranes-13-00170-f002:**
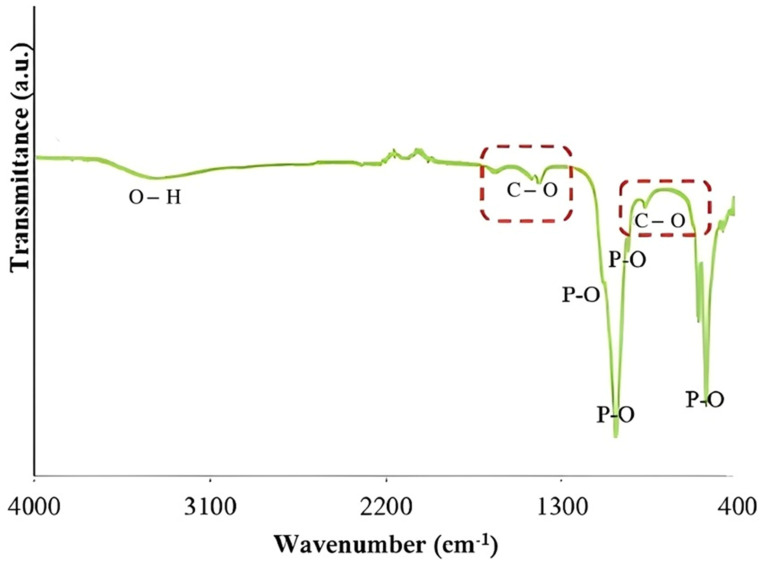
FTIR spectra of cockle shell powder.

**Figure 3 membranes-13-00170-f003:**
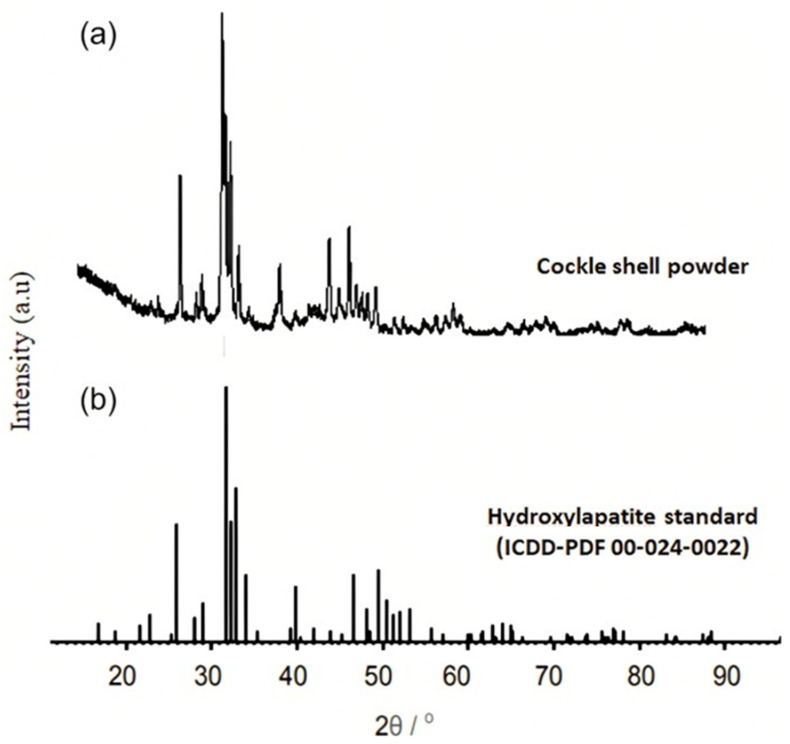
XRD diffractograms for the (**a**) cockle shell powder (**b**) standard of hydroxyapatite.

**Figure 4 membranes-13-00170-f004:**
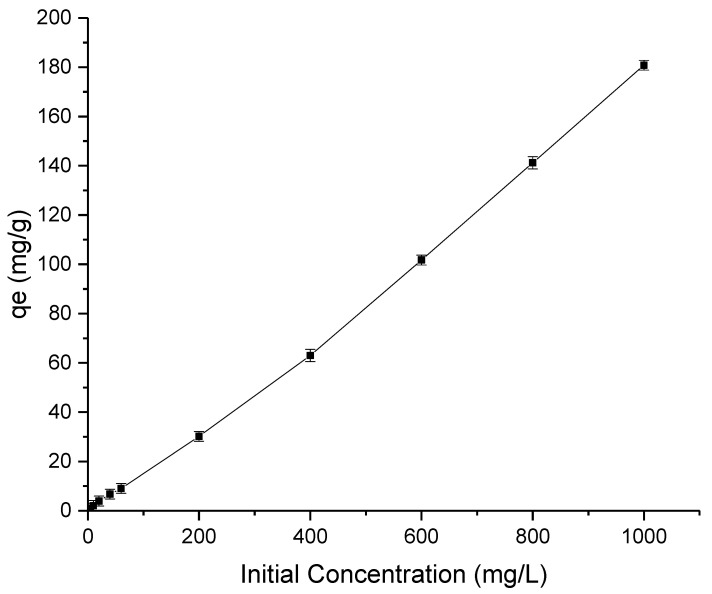
Effect of initial concentration on the lead adsorption capacity of the HAP particles.

**Figure 5 membranes-13-00170-f005:**
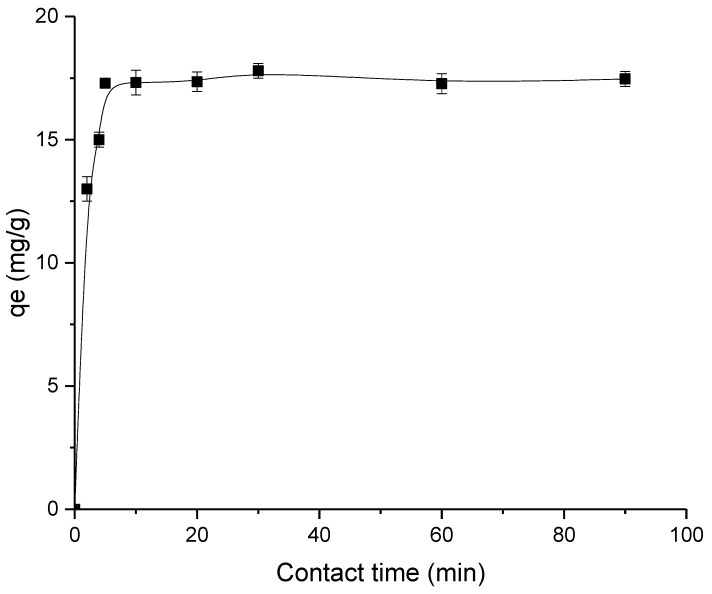
Removal of lead based on effect of contact time.

**Figure 6 membranes-13-00170-f006:**
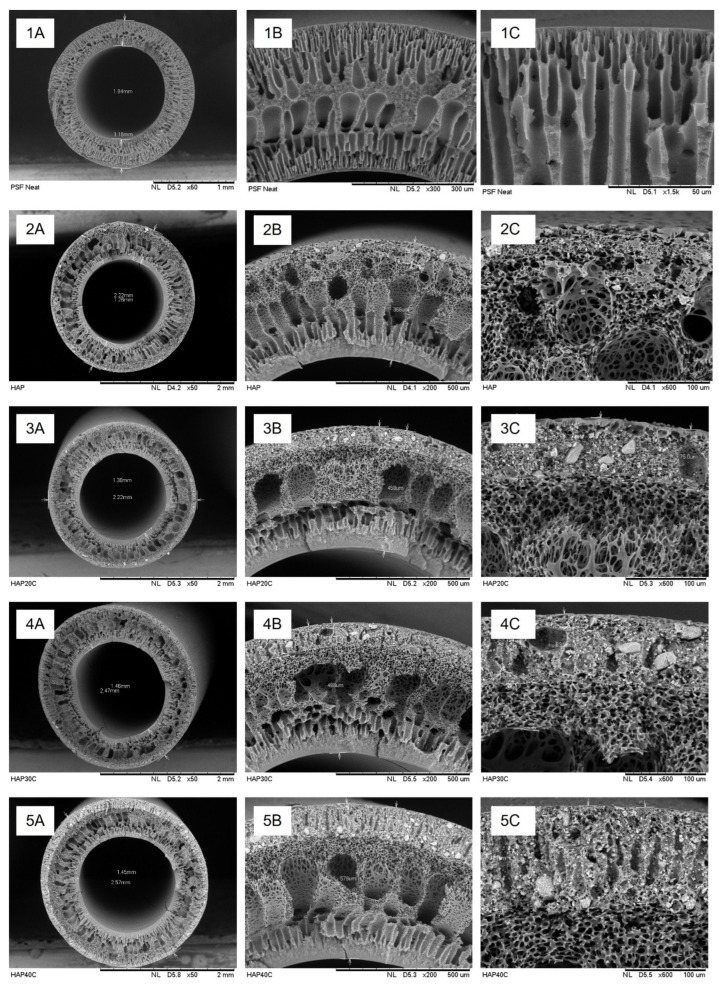
Cross-sectional SEM images of (**1**) M0, (**2**) M10, (**3**) M20, (**4**) M30, and (**5**) M40 at (**A**) 50×, (**B**) 200×, and (**C**) 600× magnification.

**Figure 7 membranes-13-00170-f007:**
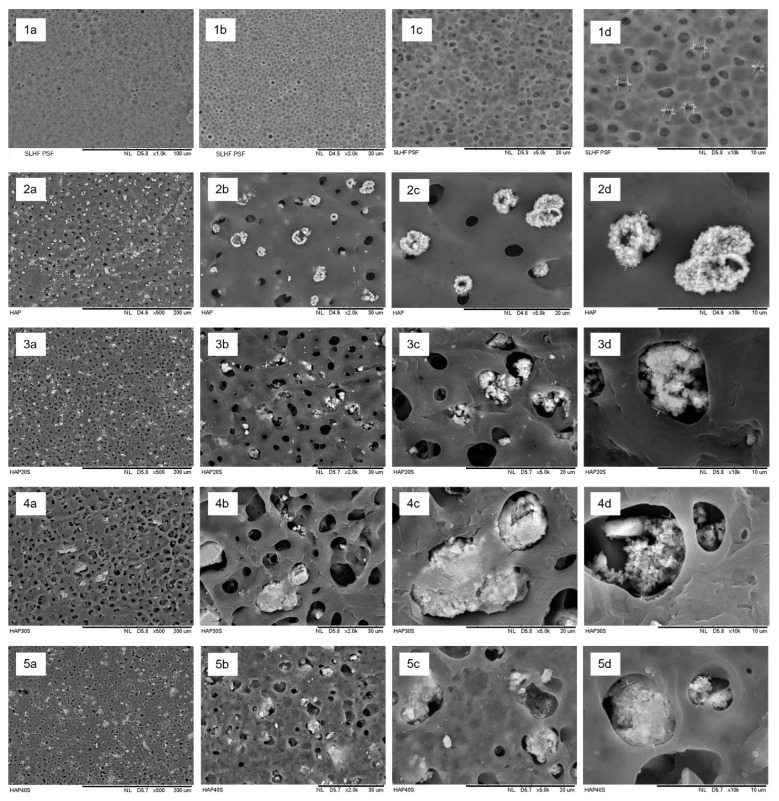
Outer surface SEM images of (**1**) M0, (**2**) M10, (**3**) M20, (**4**) M30, and (**5**) M40 at (**a**) 500×, (**b**) 2000×, (**c**) 5000× and (**d**) 10,000× magnification.

**Figure 8 membranes-13-00170-f008:**
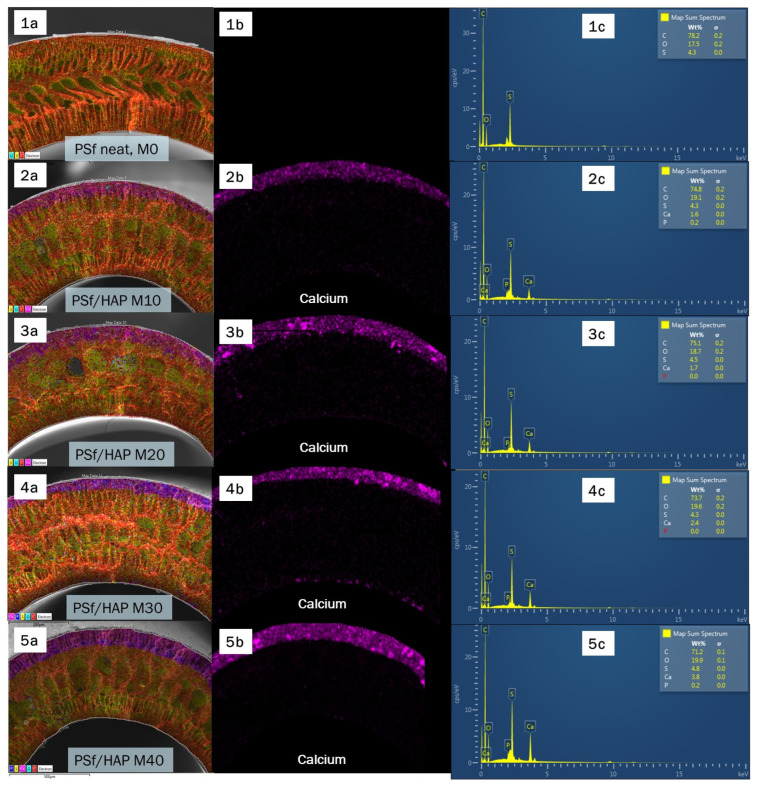
EDX mapping (**a**,**b**) and point ID (**c**) of neat PSF (**1**) and hybrid PSf/HAP DLHF (M10–M40) (**2**–**5**) at 250× (**a**) and 500× (**b**) magnification.

**Figure 9 membranes-13-00170-f009:**
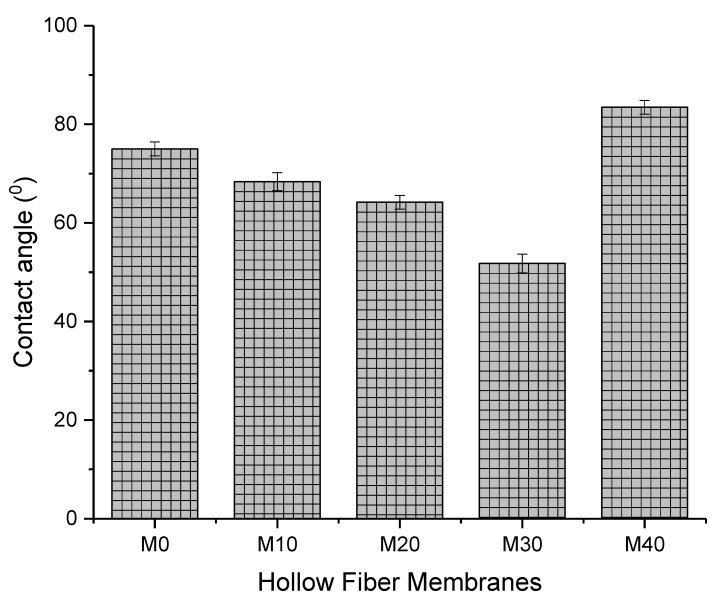
Water contact angle of the membrane outer surface. Error bars indicate standard deviations (*n* = *10*).

**Figure 10 membranes-13-00170-f010:**
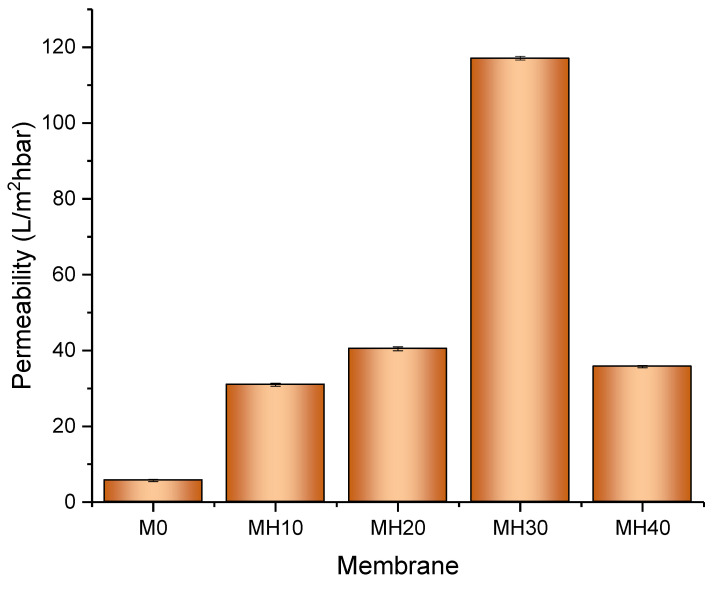
Water flux and permeability of the PSf/HAP DLHF membrane across the membrane (out-to-in cross flow) (*n* = 3).

**Figure 11 membranes-13-00170-f011:**
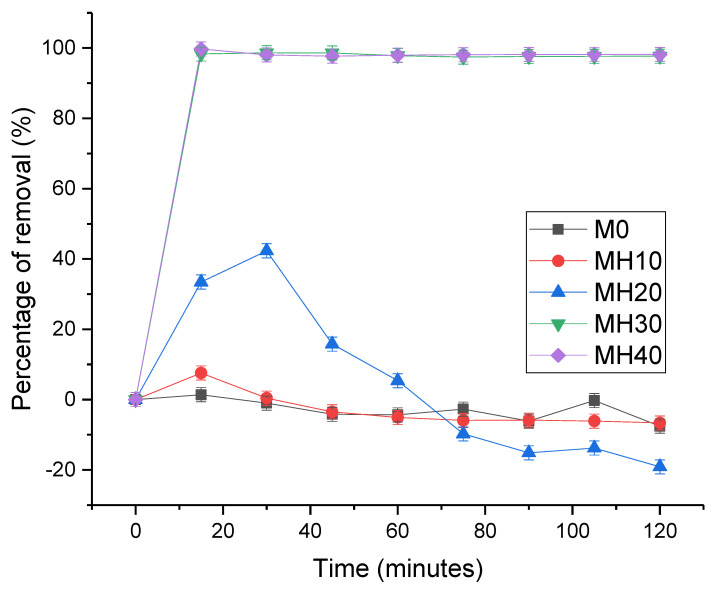
Dynamic adsorption-filtration testing for M0, MH10, MH20, MH30, MH40. The error bars indicated standard deviation at each time point. The experiment was carried out three times to ensure accurate results.

**Table 1 membranes-13-00170-t001:** Inner and outer dope composition of neat SLHF and DLHF membrane.

Membrane	Inner Dope Solution (wt%)			Outer Dope Solution (wt%)			
	PSf	PVP	NMP	PSF	PVP	HAP	NMP
M0	18	7	75	-	-	-	-
M10	18	7	75	15	5	10	70
M20	18	7	75	15	5	20	60
M30	18	7	75	15	5	30	50
M40	18	7	75	15	5	40	40

**Table 2 membranes-13-00170-t002:** Elemental components in the cockle shell powder.

	This Study	Fatimah et al. (2019)	Hajar et al. (2018)	Azis et al. (2015)
	Weight (%)			
Ca	46.8	20.1	*n/d*	22.8
C	3.0	*n/d*	9.8	*n/d*
O	48.7	67.8	68.4	*n/d*
P	0.5	11.9	8.8	12.8
Al	1.0	*n/d*	*n/d*	1.0
Co	*n/d*	*n/d*	11.9	*n/d*
K	*n/d*	*n/d*	0.8	*n/d*

**Table 3 membranes-13-00170-t003:** Langmuir and Freundlich isotherm parameters for lead adsorption on HAP.

Langmuir Model			Freundlich Model		
Q_max_ (mg/g)	*b* (L/g)	*R* ^2^	*K_F_* (mmol/g) (L/mmol)^1/n^	*n*	*R* ^2^
48.309	0.265	0.9817	10.79	0.342	0.2406

**Table 4 membranes-13-00170-t004:** Kinetics model parameters of lead molecules on HAP powder.

*C*_o_ (mg/L)	*Q*_e_ (mg/g)	Pseudo-First-Order Model		Pseudo-Second Order Model		
		*k*_1_ (1/h)	*R* ^2^	*k*_2_ (g/mgh)	*Q*_e_ (cal) (mg/g)	*R* ^2^
1000	180.8	0.59	0.7497	0.04	189.0	0.9269

## Data Availability

The data presented in this study are available upon request from the corresponding authors.

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
