# Peer review of "Hybrid Inorganic Organic PSF/Hap Dual-Layer Hollow Fibre Membrane for the Treatment of Lead Contaminated Water"

_membranes, 2023, doi:10.3390/membranes13020170_

Round 1
Reviewer 1 Report
In this paper, hydroxyapatite particles HAP was incorporated into the outer layer of polysulfone/HAP (PSf/HAP) dual-layer hollow fibre (DLHF) membrane to enhance the removal of lead from water source. But there are only brief descriptions of the results obtained, lack a deeper and more essential explanation. Some detailed comments are in the following.
1. In 2.4 Fabrication of dual-layer hollow fiber membrane, please add the full name of the first abbreviation “SLHF”.
2. HAP (Ca10(PO4)6(OH)2) is an insoluble basic phosphate, is usually insoluble in organic solvents. But in 2.4, “The HAP powder was stirred until completely dissolved in pre-calculated NMP solvent”, and there are many particles in the SEM photographs. Please determine whether HAP powder is dissolved or dispersed in NMP.
3. In 3.1, only FTIR spectra of HAP powder are given. In order to clearly understand the synthesis process of HAP powder from Waste Cockle Shell, FTIR spectra of Waste Cockle Shell should be added. What is the characteristic peak of HAP powder near 2200 cm-1?
4. In order to understand “The total pore volume and the t-plot micropore volume was 0.095 cm3 g–1 and -1.559 cm3 g–1, respectively. The negative value from t-plot micropore volume exhibited by hydroxyapatite proved that the material does not contain any micropores”, please add relevant references.
5. In the process of membrane formation via phase separation, it is generally believed that rapid phase separation results in a finger-like pore structure; delayed phase separation tends to form a spongy-like structure. However, the authors believe that due to the high viscosity of the casting solution, the phase separation slows down, the solidification in coagulating bath slow, and trapped in the new fiber, which is reason for the formation of large cavity and finger structure. Please provide strong evidence or relevant references to support this hypothesis.
6. The loss of hydrophilic additives or particles in membranes is a common problem in the membrane preparation. It is also emphasized that HAP particles are not lost from membranes. However, there is no strong data to verify the stable existence of HAP particles in the membrane.
7. As HAP concentration increases in Fig. 10, the hydrophilicity/hydrophobicity of the membrane surface appears to be consistent with the pore size of the membrane surface. The increase in surface hydrophilicity of membrane M10, M20 and M30 was attributed to the OH- and PO43- functional groups in HAP. Please add the infrared photograph and XPS element photos of the relevant membrane surface
8. In the dynamic adsorption experiment in Section 3.6, the adsorption capacity of PSf/HAP DLHF membranes with different concentrations of HAP on lead wastewater was compared. Should we consider evaluating the desorption process of the membrane after completion of adsorption saturation and adding multiple cycle adsorption tests?
Round 2
Reviewer 1 Report
The authors have revised the manuscript carefully, and it can be accepted.